# Oral Health and Hygiene Practices from Baramulla District, Jammu and Kashmir, India: A Questionnaire-Based Cross-Sectional Observational Survey

**DOI:** 10.3390/healthcare13050458

**Published:** 2025-02-20

**Authors:** Hafsa Kango, Neelu Anand Jha, Parvaiz Masoodi, Aliya Naz, Abhiroop Chowdhury

**Affiliations:** 1Independent Researcher, Roseland, Raj Bagh, Srinagar 190008, Jammu and Kashmir, India; 2Jindal School of Environment & Sustainability, O.P. Jindal Global University, Sonipat 131001, Haryana, India; achowdhury@jgu.edu.in; 3Government Medical College (GMC), and Associated Hospital, Baramulla 193101, Jammu and Kashmir, India; drasim70@gmail.com; 4Jindal School of Liberal Arts and Humanities, O.P. Jindal Global University, Sonipat 131001, Haryana, India; aliya.naz@jgu.edu.in

**Keywords:** oral health, oral hygiene practices, oral diseases, Jammu and Kashmir

## Abstract

**Background/Objectives**: Oral hygiene practices are important for good oral health and overall well-being. In this study, we surveyed people across age groups and genders to understand the oral hygiene practices of Baramulla district of the Union Territory of Jammu and Kashmir. **Methods:** 488 participants—229 males and 259 females—were randomly interviewed through a structured 20-item questionnaire. A chi-square test was used to analyze the age and gender-specific associations in oral hygiene behaviors. **Results:** The study revealed that majority of participants brushed their teeth only once a day instead of the recommended twice a day; only young cohorts showed relatively higher practice of brushing twice daily. A significantly higher proportion of males (11.69%) rarely brushed their teeth compared to females (4.28%). Tongue cleaning was poorly reported across age and genders and more than 60% of respondents did not clean their tongue at all. Almost 75% and above had made dental visits at least once. However, the visits were made primarily in cases of pain and sensitivity, indicating negligence, financial issues, or a general lack of awareness. The younger cohorts reported eating fast food more frequently, indicating risk groups. The majority of male respondents (45.88%) reported fair oral health whereas the majority of female respondents (35.29%) reported poor overall oral health. **Conclusions:** Our results show that oral hygiene practices need more improvement, and the attitude and knowledge of the residents need to be ameliorated through oral health education policy interventions.

## 1. Introduction

Oral health is an essential aspect of general health and well-being. It involves the state of the mouth, teeth, and orofacial structures. It facilitates individuals to perform essential functions like eating, breathing, and speaking, influencing psychosocial dimensions like self-confidence, social participation, and interactions, and the ability to work without discomfort or embarrassment [1]. The oral cavity is the primary pathway for entering food into the body. Therefore, oral diseases can damage the oral cavity and potentially other body parts [2].

Oral diseases are a significant global health burden, with over 3.5 billion people globally affected by oral diseases [3]. Despite being largely preventable, it causes pain, discomfort, disfigurement, and even death throughout individuals’ lifetimes [4]. Poor oral health and hygiene can cause non-communicable diseases, increase morbidity, decrease health-related quality of life (HRQoL), and incur higher medical and healthcare expenditures [5]: a total of USD 380 billion is spent on treating oral diseases, according to a WHO 2019 report [3]. Additionally, according to this report, three out of four people affected are from low and middle-income countries, indicating oral health inequality, a subject neglected in health policy discussions. This can have profound implications for achieving the Sustainable Development Goals (SDGs), particularly Goal 3, which aims to ensure good health and promote well-being for all ages [6,7]. Although SDG 3 aims to achieve good health and well-being, it does not explicitly mention any disease or healthcare system that may have implications for global health strategies [7,8]. Research shows an increased focus on this otherwise obscure public health arena, with studies being reported from different locations across the globe, such as Croatia and Mexico [9,10,11]. Abodunrin et al. 2023 opined that current research in the domain of oral health and hygiene aids compliance with other key SDGs along with SDG 3, such as SDG 1 (no poverty), SDG 4 (quality education), SDG 5 (gender equality), and SDG 13 (climate action) [8,12,13].

Oral diseases that affect various age groups are dental caries, periodontal disease, oral cancer, oro-dental trauma, noma, and birth defects like a cleft lip and palate, along with other developmental problems such as malocclusion. Dental caries or tooth decay is a significant concern and is most prevalent along with periodontal diseases, as well as oral cancers [14]. It is the most widespread oral disease among children and adolescents [15,16]. In India, 43.3% of untreated caries of deciduous teeth are prevalent in children of the one to nine age group, and 28.8% of untreated caries of permanent teeth in people above five years [17]. Another oral disease, gingivitis, is caused by the accumulation of dental biofilm and is a site-specific inflammatory condition that, if untreated, may progress to periodontitis [18]. Oral conditions are significantly higher in marginalized and vulnerable populations [19]. These diseases are largely preventable through effective oral hygiene practices and regular dental care. A significant public health concern worldwide is the growing incidences of oral cancer, particularly in communities with widespread tobacco usage, leading to an increased hospital burden and negatively affecting the survival rate [20].

Oral hygiene practices are important for good oral health. However, most people neglect oral hygiene. The use of toothpicks in place of dental floss is a common practice. Most people do not brush their teeth twice or rinse their mouths after each meal [5]. Brushing teeth at least once or preferably twice a day, along with flossing and regular visits to the dentist, is essential for good oral health [21,22]. Furthermore, good oral health practices should start early so that they become a part of long-term oral health and hygiene [23]. Understanding oral health also involves understanding the risk factors associated with it. Genetic predisposition, developmental problems, poor oral hygiene, traumatic accidents, along with modifiable risk factors such as tobacco use, alcohol consumption, stress, high sugar intake, and an unhealthy diet, which overlap with many non-communicable diseases [24], result in oral health diseases. Oral diseases also affect the ability to eat, affecting the nutritional status, and vice versa [15].

Despite advancements in dental technology and a better understanding of oral diseases, significant disparities in dental health and access to care persist across various demographics. These disparities are influenced by income, race or ethnicity, geographic location, and education. Implementing good oral health and hygiene policies involves understanding the desired population’s knowledge, behavior, attitudes, and practices. There is sufficient literature on oral health from various parts of the country [25,26,27,28,29], but, to the best of our knowledge, there are very few articles from Jammu and Kashmir [30,31,32] and none from the Baramulla district. Baramulla is the fourth most populous town in Jammu and Kashmir, with a population of 1,008,039 residents (Census of India, 2011) of which 53.03% are males and 46.95% are females. It is known for its geographical significance and historical legacy [33,34]. It is a remote area in the northwest direction located on the banks of the Jhelum River. It is an Aspirational District, 1 of the 112 districts from the entire country, chosen by the Government of India under the Aspirational Districts Program to quickly transform the most under-developed districts of the country through cooperative and competitive federalism by focusing on five socio-economic themes such as health and nutrition, education, agriculture and water resources, financial inclusion, and skill development [35]. In recent years, it has seen improvements in infrastructure, including enhanced road networks and the establishment of new educational institutions that aim to improve the quality of life for its residents [36]. A systematic literature search with keywords, (“oral health” OR “oral hygiene” OR “oral hygiene practices” OR “oral hygiene behavior”) AND (“knowledge”) AND (“attitude”) and (“practices”) showed few research articles from the Kashmir region and none from Baramulla. According to the Dental Council of India, Jammu and Kashmir report, more than 85% of the children and 95–100% of the adult population of Jammu and Kashmir suffer from periodontal diseases [37], and there is a dire necessity for policies and manpower development. Even though Baramulla has a tertiary hospital and the availability of healthcare services, a lack of real-time data on oral and dental status is required for the strategic planning of improving oral healthcare services. This study was conducted to fill the literature gap regarding Baramulla’s oral health status and to understand the practices regarding the oral health and hygiene of its residents. This study can be used as a baseline for understanding the oral health and hygiene of people from varied socio-economic backgrounds in the Baramulla district. This kind of study can be useful for identifying risk factors leading to oral diseases and to help in strategizing oral care prevention through policy interventions.

## 2. Methods

A questionnaire-based cross-sectional study was done from September 2023 to November 2023 at the Dental Department of the Government Medical College and its associated hospital, Baramulla (34.1990° N, 74.3499° E), to understand the oral health and hygiene practices of people across various age groups of Baramulla, a district in the Kashmir division of the Indian Union Territory Jammu and Kashmir. The Government Medical College and its associated hospital in Baramulla, a tertiary hospital in the district, provide specialized medical care. The Baramulla district has 10 medical blocks, and people from these medical blocks, as well as nearby villages from other districts, visit this hospital to access affordable healthcare.

### 2.1. Study Subjects and Demography 

The study was conducted on the hospital premises during the working hours of the Outpatient Department (OPD), i.e., from 10:00 a.m. until 02:00 p.m. Participation in the study was voluntary, with proper ethics maintained. People present on the hospital premises during the study period who agreed to participate in the survey were included in the study. Those who denied being interviewed were excluded from the study. An informed verbal consent was sought before the study following the principle of Helsinki for human studies [38]. For minors, consent was sought from their parents/guardians. They were briefed and assured that all the information they shared would be confidential and used only for scientific study. Unique identification codes maintained the confidentiality and anonymity of the subjects. The study was conducted after seeking approval from the Office of the Medical Superintendent, Government Medical College, and its associated hospital, Baramulla (AH-GMCB/2023/907; 8 September 2023). A total of 488 participants were part of the study, of which 229 (46.93%) were males and 259 (53.07%) were females. The sample size calculation was performed using the sample size calculator from the Department of Quantitative Health Sciences, Cleaveland Clinic (https://riskcalc.org/samplesize/ (accessed on 5 August 2023)). The total population of Baramulla as reported by the Census of India, 2011, was 1,008,039. Our sample size of 488 was conveniently above the minimum of 384 participants required to reach a 95% confidence level and a 5% margin of error. The participants were from across ages. We divided our participants into the following age categories: 1–10 years, 11–20, 21–30, 31–40, 41–50, 51–60, and above 60. We categorized everyone above the age of 60 years in one group as we had very few respondents above 61 years of age. Figure 1 indicates the native locations of the participants from the district.

### 2.2. Assessment Through Questionnaire

A self-structured questionnaire consisting of 20 items was prepared to assess the basic oral health practices of the study subjects. The questionnaire is based on the World Health Organization’s (WHO) Oral Health Survey Manual [39]. Out of the 20 questions, 17 were close-ended, and 3 were open-ended. The researcher read out the questions to the respondents and filled out their responses. It took 7–10 min per subject to complete the questionnaire. However, in the case of participants under 10, it took slightly more time to complete the questionnaire since the answers were provided by the parents, but some young participants also interacted during the session. The questionnaire collected basic demographic information related to the participant’s name, age, sex, occupation, address, education, and socio-economic status, and self-assessment of one’s current oral health status and practices such as teeth cleaning, brushing, and oral health problems and visits to dentists. We determined the socio-economic status of the people based on the type of Ration Card they had. A Ration Card is a Public Distribution System document in India issued by the state governments, which divides households into different priority groups to provide food grains at subsidized rates [40]. According to this system, the Below Poverty Line (BPL) card is issued to households living below the poverty line, and the Above Poverty Line (APL) is issued to households living above the poverty line. Priority Households (PHH) and Antyodaya Anna Yojana (AAY) are special beneficiary categories issued to households based on specific criteria of the state governments. Along with these questions, information about eating fast food and indulging in tobacco products, which impact oral health, was asked (refer to Appendix B).

### 2.3. Statistical Analysis

We collected the data and then recorded, compiled, and entered it into the MS Office Excel sheet. This Excel sheet was used to perform descriptive statistical analysis and chi-square tests. We used chi-square tests to analyze the age and gender-specific associations in oral hygiene behaviors using the Past 5.0.2 software. For all chi-square tests, the significance level was set at 0.05.

## 3. Results

Socio-demographic characteristics: A total of 488 residents of Baramulla participated in the study. Of the total participants, 46.93% (229) were male and 53. 07% (259) were females. The youngest respondent was 4 years old, and the oldest was 80. We categorized them into the following age brackets: 1–10 years, 11–20, 21–30, 31–40, 41–50, 51–60, and above 60 years old. Most of our respondents were young individuals. We had 146 of the total 488, i.e., 29.92%, from the age bracket of 21–30 years, followed by 21.72% of participants from the 31–40 age group and 16.80% of 11–20 years. We had very few participants above 51 years of age: 3.89% in the 51–60 age group and 2.66% above the 61 years of age bracket (see Table 1) as they were reluctant to participate in the survey. The Baramulla district has 10 medical blocks. The residences of the study participants were also divided into the 10 Medical Blocks and presented in Appendix A. We had fair participation from both males and females in our study (Table 2), except in the age group of 51–60 years, where we had 84.21% male participation and around 15.79% female participation.

Of the people interviewed, 41.39% belonged to the BPL category, 42.62% belonged to the APL, and 15.98% belonged to the PHH and AAY category, indicating that we had participation from different socio-economic strata (socio-economic status, Table 1 and Appendix A). Most of our respondents had attained basic education, and only 8.81% were non-literate (Appendix A).

### 3.1. Oral Hygiene Behaviour

The results show that most respondents brushed their teeth once a day. The chi-square test revealed that there was a significant association of tooth brushing frequency across age *p* < 0.001, Table 3a) and gender (*p* = 0.02, Table 3b). The highest percentage of the respondents in each age group brushed their teeth once a day (Table 3a). In younger age groups, i.e., 11–20, 21–30, and 31–40, the second highest percentage of the residents brushed their teeth twice daily (26.83% in the 11–20 age bracket, 27.08% in the 21–30 age bracket, and 20.75% in the 31–40 age bracket). In the higher age group, a larger proportion of residents either brushed their teeth alternately or seldom or never (Table 3a). 64.98% of females brushed their teeth once, compared to 58.87% of males. Additionally, 11.69% of males brushed their teeth rarely, as opposed to only 4.98% of female respondents (Table 3b). Most of the respondents across age groups and genders brushed their teeth in the morning as opposed to brushing their teeth at night (Table 4a,b). The chi-square test revealed an association of age *p* < 0.001) but not of gender (*p* = 0.25) on the timing of tooth-brushing. There was a decrease in the proportion of respondents brushing their teeth with increasing age (see Table 4b). All the respondents reported using toothbrushes and toothpaste as oral hygiene aids.

The residents of Baramulla showed a very poor frequency of cleaning their tongues. Over 60% of respondents in each age group did not clean their tongues, and only 30% or less cleaned their tongues daily (Table 5a,b). There was an association of age but not of gender on the tongue cleaning behavior (*p* < 0.001; *p* = 0.16, respectively, chi-square test). The younger age brackets of 1–10 and 11–20 had the highest proportion of individuals not cleaning their tongue (95.16% and 82.93%, respectively), with a slight improvement in tongue-cleaning habits in higher age groups (see Table 5a). There was an association of age but not of gender in the usage of mouthwash or other cleaning aids (*p* < 0.001; *p* = 0.211, respectively, chi-square test, Table 6a,b). More than 60% of people across age groups did not use mouthwash or any other oral aid to keep their mouth clean (Table 6a,b). The use of mouthwash and any other cleaning aid declined with age, with more than 80% of people above 50 years not using any aid to keep their mouth clean except in age group 1–10, where the percentage was above 90% (Table 6a).

### 3.2. Dental Visits

All the people above 61 years of age and more than 85% of people above the age of 21 had visited the dentist at least once (Table 7a). Most of the people primarily visited if they felt pain, sensitivity, or discomfort in their teeth. The chi-square analysis revealed an association of age (*p* = 0.02) but not of gender (*p* = 0.96) in the frequency of dental visits (Table 7a,b). Apart from this, many respondents had additional oral/dental issues for which they visited the doctor, such as filling, root canal therapy (RCT), tooth extraction or multiple teeth extraction, medication, ortho, scaling, splinting, and yellow teeth (Table 7c).

### 3.3. Oral Health-Related Dietary Behaviour

The chi-square analysis revealed an association with age (*p* < 0.001) but not with gender (*p* = 0.17) in the frequency of eating fast food (Table 8a,b). The highest proportion of the young respondents (43.55% in 1–10, 43.90% in 11–20, and 34.25% in 21–30 age bracket) had fast food daily, whereas the highest proportion of oldest participants, i.e., above 60 years of age, rarely had fast food (see Table 8a).

### 3.4. Tobacco Consumption

More than 70% of people across all age groups did not report consuming tobacco. Most of the respondents who did consume tobacco did not reveal how many cigarettes they had in a day or for how long they had been using tobacco or cigarettes. Use of tobacco had an association with age (*p* < 0.001; chi-square test, Table 9a). The percentage of respondents who reported consuming tobacco increased with increasing age brackets (see Table 9a). There was no association of gender t *p* = 0.28; chi-square test, Table 9b).

### 3.5. Self-Rating of Overall Oral Health

Very few participants reported excellent or very good oral health. Chi-square analysis showed an association of gender (*p* < 0.001) but not of age (*p* = 0.19) on the self-reported overall oral health (Table 10a,b). The majority of male respondents (45.88%) reported fair oral health, whereas the majority of female respondents (35.29%) reported poor overall oral health (Table 10b).

## 4. Discussion

This study aimed to examine the oral health and hygiene practices of the people of the Baramulla district in the Indian Union Territory of Jammu and Kashmir. This is the first account of the oral hygiene practices from the Baramulla district, a gateway to the Kashmir Valley. Between the Pir Pranjal Mountain range in the northwest and the fertile plains of the Indus and Ganges in the south, Baramulla is fast catching up with health, education, and infrastructure development [36,41]. Such baseline studies are important for identifying risk groups and risk factors leading to oral diseases and to help strategize oral care prevention through policy interventions.

Cleaning teeth is an essential part of oral hygiene practice. It is an established method of removing dental plaque, stains, and food deposits from the teeth, thereby preventing dental caries and periodontal diseases [15]. Our study shows that the highest proportion of the residents in each age group brushed their teeth only once. It does not meet the oral recommendation of brushing the teeth twice daily [42,43]. Studies on tooth brushing frequency within the Indian population reveal similar trends. A study from Kancheepuram District, Tamil Nadu, reported that 56% of the urban population brushed once daily, while 44% brushed twice daily, whereas 86% of the rural population brushed once daily, and only 13% brushed twice daily [44]. A study focusing on oral health awareness in Chennai found that 37.62% of participants brushed twice daily, while the remaining participants brushed only once a day [45]. A recent report indicates that approximately 45% of Indians brush their teeth twice daily, which is notably lower than several countries [46]. For example, the prevalence of twice-a-day tooth brushing is seen in a higher proportion of the cohorts studied: 51% in the Australian, 75% in the English and 85% in the Swedish population [47,48,49]. Interestingly, in the younger cohort (from 11–40 years), the second highest proportion of people brushed their teeth twice. This indicates that the young population is possibly aware of good oral practices due to the oral health education programs being implemented in new education policies and greater awareness and heightened body image in the new digital era. These data are important in affirming the role of implementing oral education at school levels. Young children develop reflexes toward general hygiene practices and overall attitude toward health early in childhood [22]. Therefore, oral health education should be integral to the school curriculum. Data from countries like Australia and New Zealand, which have excellent oral health education programs, show decreased dental caries among school children [22]. Our study shows a higher proportion of elderly people either brushed their teeth alternately or never. This is similar to studies that show older populations with poor oral hygiene and health indicating oral frailty [50,51,52]. The study on 45–74-year-old men from Northeast Poland found that only 53.2% of respondents brushed their teeth once a day and only 21% used additional measures to maintain oral hygiene [50]. We found a relatively higher number of males rarely or seldom brushing their teeth as compared to females. A similar study from Japan showed that the rate of good oral behavior, including tooth brushing, was significantly higher among females than among males [53].

Cleaning of the tongue enhances oral hygiene [54]. However, our study population of Baramulla showed poor tongue-cleaning behavior across age groups, similar to previous studies [55,56] as well as less usage of mouthwash or any other oral aid to keep their mouth clean. On the contrary, a study conducted at a dental college in Jamnagar found that 67.2% of patients cleaned their tongues, with a higher prevalence among males (68.9%) compared to females (64.8%) [57]. A dental health survey in Scotland revealed that most of the participants (38.1%) had never used mouthwash, similar to our survey result [58]. We observed a general reluctance and hesitancy among respondents regarding personal oral hygiene questions. This may be due to unwillingness to seek oral care due to negligence, financial issues, geographic isolation, and time-consuming visit procedures [59,60]. The data show that most of the population has made visits to the dental hospital or the dentist, but the primary reasons for visiting have been pain and sensitivity. The common practice that the respondents conveyed was seeking medications with no intention of treatment or removal of the tooth because of financial constraints, dental anxiety, fear, length of procedures, long waiting hours, and more visits. Upon medication, if the pain subsided, most people did not follow up with a dental checkup.

Fast-food consumption patterns varied between the age groups, with the younger population having fast food either daily or several days a week as opposed to older age groups that preferred not to consume fast food. This is similar to reports indicating age as an important determinant in dietary choices [61]. Consumption of fast food negatively affects oral health [62]. The oral health of adolescents and young people is usually bad since they have poor dietary choices and an overconsumption of sugary drinks, snacks, and fast foods. They also tend to disregard their guardian’s advice and may not be inclined towards good hygiene [15].

Our study reported low consumption of tobacco. Collecting data on tobacco usage is challenging owing to the sensitive personal, informal, and social stigma associated with it. The majority of male respondents reported fair oral health as opposed to females who reported poor oral health. This is contrary to several other studies where women tend to report a higher perception of their oral health status than men. They also viewed their oral health positively, prioritized oral health care, and took great interest in personal appearance [63,64,65]. However, some reports indicate that women have a lower self-assessment of oral health [66]. This might be because of lower self-esteem in women due to societal, psychological, or economic factors [67].

The study should be treated as baseline data for investigating the oral health and hygiene behavior of the Baramulla district. The study has several limitations that we would like to address in the future. This study could not establish the knowledge and attitude of the people towards oral health. The socio-economic and educational levels impacting oral health were not examined. Furthermore, a longitudinal study tracking changes in practices concerning oral health after imparting knowledge regarding oral health would be recommended. The cause-and-effect relationship between oral hygiene practices and oral health is warranted. The present study should have ideally included a pilot study with 15–20 respondents as per the WHO recommendations, and the responses should have been validated with standard statistical measures such as Cronbach’s alpha. Additionally, an objective assessment of oral health by expert medical practitioners, along with self-reported overall oral health, would have given more confidence to the dataset.

## 5. Conclusions

Overall, our study indicates that the oral hygiene practices of the Baramulla district require significant improvement. The frequency of oral cleaning falls below recommended standards, with a noticeable decline with increasing age, reflecting a general lack of awareness and attention to oral health. This study contributes valuable data on oral hygiene behaviors and ongoing oral health challenges in the 21st century. Integrating oral health into the United Nations’ Sustainable Development Goals (SDGs), where it remains largely overlooked, would be a crucial step toward policy-level interventions. Enhancing people’s knowledge and attitudes through targeted oral health education programs, particularly school-based initiatives, outreach and awareness campaigns, preventive dental services, and healthy nutrition policies can help establish lifelong good hygiene behaviors, especially among vulnerable groups, such as children, the elderly, and marginalized low-income populations.

## Figures and Tables

**Figure 1 healthcare-13-00458-f001:**
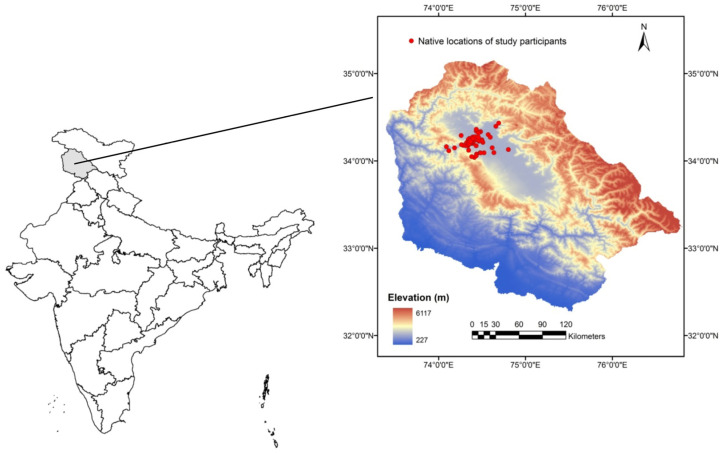
Map representing the native locations of the respondents included in the study. All the respondents belonged to the Baramulla district of Jammu and Kashmir, India. The map was prepared in Arc GIS 9.3 software using the Digital Elevation Model (DEM) of the study area as a base.

**Table 1 healthcare-13-00458-t001:** Socio-demographic characteristics of the study participants.

Total Respondents for the Study: 488	
**Socio-Demographic Variables**	
**Gender**	
Male	229 (46.93%)
Female	259 (53.07%)
**Age** (**years**)	
1–10	62 (12.70%)
11–20	82 (16.80%)
21–30	146 (29.92%)
31–40	106 (21.72%)
41–50	60 (12.30%)
51–60	19 (3,89%)
61 and above	13 (2.66%)
**Socio-economic status**	
Below Poverty Line (BPL)	202 (41.39%)
Above Poverty Line (APL)	208 (42.62%)
Priority Household (PHH)/Antyodaya Anna Yojana (AAY)	78 (15.98%)
**Education**	
Primary School	57 (11.68%)
Middle School	82 (16.80%)
High School	138 (28.28%)
Undergraduate/Diploma	102 (20.90%)
Postgraduate/Ph.D./Higher	52 (10.66%)
Non-literate/Informal/Maqtab/Others	43 (8.81%)
Did not start schooling	10 (2.05%)
Did not respond	4 (0.82%)

**Table 2 healthcare-13-00458-t002:** Table depicting the distribution of male and female participants in various age groups.

Age (Years)	Total Participants	Male Participants	Female Participants
1–10	62	25 (40.32%)	37 (59.67%)
11–20	82	34 (41.46%)	48 (58.54%)
21–30	146	61 (41.78%)	85 (58.22%)
31–40	106	50 (47.17%)	56 (52.83%)
41–50	60	35 (58.33%)	25 (41.66%)
51–60	19	16 (84.21%)	3 (15.79%)
61 and above	13	8 (61.54%)	5 (38.48%)

**Table 3 healthcare-13-00458-t003:** (**a**). Prevalence of tooth-brushing behavior among the study participants across various age groups. (**b**). Table showing male and female differences in the prevalence of tooth-brushing behavior across various age groups. The notations in the brackets represent the percentage of the population.

(**a**)								
**Age** (**Years**)	**Frequency of Tooth-Brushing in a Day**							
	**Twice**		**Once**		**Alternate**		**Rarely/Seldom/Never**	
1–10	1 (1.61%)		42 (67.74%)		13 (20.97%)		6 (9. 68%)	
11–20	22 (26.83%)		56 (68.29%)		1 (1.21%)		3 (3.66%)	
21–30	39 (27.08%)		83 (57.64%)		14 (9.72%)		8 (5.56%)	
31–40	22 (20.75%)		67 (63.20%)		10 (9.43%)		7 (6.60%)	
41–50	7 (11.67%)		39 (65%)		7 (11.67%)		7 (11.67%)	
51–60	1 (5.26%)		8 (42.11%)		6 (31.58%)		4 (21.05%)	
61 and above	1 (7.69%)		6 (46.15%)		3 (23.07%)		3 (23.07%)	
χ^2^ _(18, 488)_ = 59.59, *p* < 0.001.								
(**b**)								
**Age** (**Years**)	**Twice**		**Once**		**Alternate**		**Rarely/Seldom/Never**	
	**Male**	**Female**	**Male**	**Female**	**Male**	**Female**	**Male**	**Female**
1–10	0	1 (2.70)	16 (64.00)	26 (70.27)	6 (24.00)	7 (18.91)	3 (12.00)	3 (8.10)
11–20	10 (31.25)	12 (24.00)	18 (56.25)	38 (76.00)	1 (3.13)	0	3 (9.38)	0
21–30	16 (26.23)	23 (26.44)	36 (59.02)	47 (54.02)	3 (4.92)	11 (12.64)	6 (9.84)	2 (2.30)
31–40	12 (23.08)	10 (18.52)	33 (63.46)	34 (62.96)	4 (7.69)	6 (11.11)	3 (5.77)	4 (7.41)
41–50	4 (11.76)	3 (11.540)	24 (70.59)	15 (57.69)	2 (5.88)	5 (19.23)	4 (11.76)	3 (11.54)
51–60	1 (6.25)	0	5 (31.25)	3 (100.00)	6 (37.50)	0	4 (25.00)	0
61 and above	1 (12.50)	0	3 (37.50)	3 (60.00)	2 (25.00)	1 (20.00)	2 (25.00)	1 (20.00)
χ^2^ _(3, 488)_ = 9.49, *p* = 0.02.								

**Table 4 healthcare-13-00458-t004:** (**a**). Time of tooth-brushing during the 24-h day of the study participants across various age groups. (**b**). Table depicting male and female differences in tooth-brushing time during the 24-h day among the study participants across various age groups. The notations in the brackets represent the percentage of the population.

(**a**)								
**Age** (**Years**)	**Time of Tooth-Brushing During the 24-h Day**							
	**Morning**		**Morning + Night**		**Night**		**No Response**	
1–10	44 (70.97%)		1 (1.61%)		1 (1.61%)		16 (25.81%)	
11–20	51 (63.75%)		20 (25.00%)		5 (6.25%)		4 (5.00%)	
21–30	79 (53.74%)		39 (26.53%)		6 (4.08%)		23 (15.64%)	
31–40	61 (57.55%)		23 (21.70%)		5 (4.72%)		17 (16.04%)	
41–50	33 (55.00%)		7 (11.67%)		6 (10.00%)		14 (23.33%)	
51–60	7 (36.84%)		1 (5.26%)		1 (5.26%)		10 (52.63%)	
61 and above	6 (46.15%)		1 (7.69%)		1 (7.69%)		5 (38.46%)	
χ^2^ _(18, 488)_ = 56.83, *p* < 0.001.								
(**b**)								
**Age** (**Years**)	**Morning**		**Morning + Night**		**Night**		**No Response**	
	**Male**	**Female**	**Male**	**Female**	**Male**	**Female**	**Male**	**Female**
1–10	16 (64.00)	28 (75.68)	0	1 (2.70)	0	1 (2.70)	9 (36)	7 (18.92)
11–20	17 (56.67)	34 (68.00)	8 (26.67)	12 (24.00)	1 (3.33)	4 (8.00)	4 (13.33)	0
21–30	35 (55.56)	44 (52.38)	16 (25.40)	23 (27.38)	2 (3.17)	4 (4.76)	10 (15.87)	13 (15.48)
31–40	30 (58.82)	31 (56.36)	12 (23.53)	11 (20.00)	2 (3.92)	3 (5.45)	7 (13.73)	10 (18.18)
41–50	19 (54.29)	14 (56.00)	4 (11.43)	3 (12.00)	5 (14.92)	1 (4.00)	7 (20.00)	7 (28.00)
51–60	5 (31.25)	2 (66.67)	1 (6.25)	0	0	1 (33.33)	10 (62.50)	0
61 and above	4 (50)	2 (40.00)	1 (12.50)	0	0	1 (20.00)	3 (37.50)	2 (40.00)
χ^2^ _(3, 488)_ = 4.09, *p* = 0.25.								

**Table 5 healthcare-13-00458-t005:** (**a**). Frequency of tongue-cleaning behavior among the study participants across various age groups. (**b**). Male and female differences in tongue-cleaning behavior among the study participants across various age groups. The notations in the brackets represent the percentage of the population.

(**a**)								
**Age** (**Years**)	**Frequency of Cleaning the Tongue**							
	**Yes (Daily)**		**Yes (Sometimes)**		**No**		**No Response**	
1–10	1 (1.61%)		1 (1.61%)		59 (95.16%)		1 (1.61%)	
11–20	14 (17.07%)				68 (82.93%)			
21–30	39 (26.71%)		7 (4.79%)		99 (67.80%)		1 (0.68%)	
31–40	25 (23.58%)				81 (76.42%)			
41–50	13 (21.67%)				47 (78.33%)			
51–60	2 (10.53%)				17 (89.47%)			
61 and above	4 (30.77%)				9 (69.23%)			
χ^2^ _(6, 488)_ = 21.42, *p* < 0.001.								
(**b**)								
**Age** (**Years**)	**Yes** (**Daily**)		**Yes** (**Sometimes**)		**No**		**No Response**	
	**Male**	**Female**	**Male**	**Female**	**Male**	**Female**	**Male**	**Female**
1–10	0	1 (2.70)	0	1 (2.70)	24 (96.00)	35 (94.59)	1 (4.00)	0
11–20	5 (14.71)	9 (18.00)			29 (85.29)	41 (82.00)		
21–30	11 (17.74)	28 (33.33)	3 (4.84)	4 (4.76)	47 (75.81)	52 (61.90)	1 (1.61)	0
31–40	15 (30.00)	10 (17.86)			35 (70.00)	46 (82.14)		
41–50	5 (14.29)	8 (32.00)			30 (85.71)	17 (68.00)		
51–60	2 (12.50)	0 (32.0)			14 (87.50)	3 (100.00)		
61 and above	1 (14.29)	3 (50.00)			6 (85.71)	3 (50.00)		
χ^2^ _(1, 488)_ = 2.01, *p* = 0.16.								

**Table 6 healthcare-13-00458-t006:** (**a**). Use of mouthwash or any other aid for keeping mouth clean among the study participants across various age groups. (**b**). Table showing the male and female differences among the study participants in the use of mouthwash or any other aid for keeping the mouth clean across the age groups. The notations in the brackets represent the percentage of the population.

(**a**)						
**Age** (**Years**)	**Yes**		**No**		**No Response**	
1–10	2 (3.23%)		58 (93.54%)		2 (3.23%)	
11–20	19 (23.17%)		55 (67.07%)		8 (9.76%)	
21–30	47 (32.41%)		90 (62.07%)		8 (5.52%)	
31–40	32 (30.19%)		70 (66.04%)		4 (3.77%)	
41–50	7 (14.89%)		37 (78.72%)		3 (6.38%)	
51–60	2 (10.52%)		17 (89.47%)			
61 and above	2 (15.38%)		11 (84.62%)			
χ^2^ _(12, 488)_ = 35.15, *p* < 0.001.						
(**b**)						
**Age** (**Years**)	**Yes**		**No**		**No Response**	
	**Male**	**Female**	**Male**	**Female**	**Male**	**Female**
1–10	0	2 (5.41)	24 (96.00)	34 (91.89)	1 (4.00)	1 (2.70)
11–20	8 (25.00)	11 (22.00)	21 (65.63)	34 (68.00)	3 (9.38)	5 (10)
21–30	14 (23.33)	33 (37.93)	42 (70.00)	48 (55.17)	4 (6.67)	6 (6.90)
31–40	15 (25.00)	17 (30.36)	44 (73.33)	36 (64.29)	1 (1.67)	3 (5.36)
41–50	5 (17.86)	2 (10.53)	21 (75.00)	16 (84.21)	2 (7.14)	1 (5.26)
51–60	2 (12.50)	0	14 (87.50)	3 (100.00)		
61 and above	1 (14.29)	1 (16.67)	6 (85.71)	5 (83.33)		
χ^2^ _(2, 488)_ = 3.10, *p* = 0.211.						

**Table 7 healthcare-13-00458-t007:** (**a**). Records of visits to dental hospitals/dentists by the study participants across various age groups. (**b**). Male and female differences in visits to dental hospitals/dentists among the study participants across various age groups. The notations in the brackets represent the percentage of the population. (**c**). Frequency of oral issues faced by the participants (apart from pain and sensitivity) for which they have/had visited the hospital (some respondents had more than one oral issue). The notations in the brackets represent the percentage of the population.

(**a**)											
**Age** (**Years**)	**Yes**						**No**				
1–10	48 (77.42%)						14 (25.80%)				
11–20	63 (76.83%)						19 (23.17%)				
21–30	126 (90.00%)						18 (10.00%)				
31–40	93 (87.74%)						13 (12.26%)				
41–50	56 (93.33%)						4 (6.67%)				
51–60	17 (89.47%)						2 (10.53%)				
61 and above	13 (100.00%)										
χ^2^ _(6, 488)_ = 14.62, *p* = 0.02.											
(**b**)											
**Age** (**Years**)	**Yes**						**No**				
	**Male**			**Female**			**Male**			**Female**	
1–10	16 (64.00)			31 (83.78)			9 (36.00)			6 (16.22)	
11–20	27 (84.38)			26 (65.00)			5 (15.63)			14 (35.00)	
21–30	53 (88.33)			73 (86.90)			7 (11.67)			11 (13.10)	
31–40	41 (82.00)			52 (92.86)			9 (18.00)			4 (7.14)	
41–50	32 (91.43)			24 (96.00)			3 (8.57)			1 (4.00)	
51–60	15 (93.75)			2 (66.67)			1 (6.25)			1 (33.33)	
61 and above	8 (50.00)			5 (100.00)							
χ^2^ _(1, 488)_ = 0.003, *p* = 0.96.											
(**c**)											
**Age** (**Years**)	**Fillings**	**RCT**	**Extraction**	**Multiple Extraction**	**Medication**	**Ortho**	**Scaling**	**Yellow Teeth**	**Splinting**	**Never Visited**	**No Response**
1–10											
11–20	22 (21.36)	20 (19.42)	13 (12.62)	3 (2.91)	3 (2.91)	1 (0.97)	1 (0.97)	1 (0.97)	1 (0.97)	1 (0.97)	37 (35.92)
21–30	1 (20.00)	1 (20.00)	2 (40.00)					1 (20.00)			
31–40	24 (15.56)	20 (13.07)	45 (29.41)	18 (11.76)	5 (3.27)	4 (2.61)	4 (2.61)				33 (21.57)
41–50	18 (18.75)	15 (15.63)	29 (30.21)	17 (17.71)	2 (2.08)		4 (4.17)				11 (11.46)
51–60	4 (13.79)	6 (20.69)	5 (17.24)	6 (20.69)		2 (6.90)					6 (20.69)
61 and above	2 (11.76)	4 (23.53)	7 (41.18)	2 (11.76)							2 (11.76)

**Table 8 healthcare-13-00458-t008:** (**a**). Frequency of eating fast food among the study participants across various age groups. (**b**). Table showing male and female differences in frequency of eating fast food across various age groups. The notations in the brackets represent the percentage of the population.

(**a**)												
**Age** (**Years**)	**Daily**		**Several Times a Week**		**Once a Week**		**Once in a While**		**Rarely**		**Never**	
1–10	27 (43.55%)		8 (12.90%)		2 (3.23%)		3 (4.84%)		5 (8.06%)		15 (24.19%)	
11–20	36 (43.90%)		10 (12.20%)		14 (17.07%)		11 (13.41%)		7 (8.54%)		4 (4.88%)	
21–30	50 (34.25%)		19 (13.01%)		29 (19.86%)		31 (21.23%)		9 (6.16%)		5 (3.42%)	
31–40	19 (17.92%)		21 (19.81%)		18 (16.98%)		22 (20.75%)		13 (12.26%)		12 (11.32%)	
41–50	10 (16.67%)		12 (20.00%)		11 (18.33%)		13 (21.67%)		8 (13.33%)		5 (8.33%)	
51–60	6 (31.58%)		2 (10.52%)		2 (3.33%)		2 (3.33%)		3 (15.79%)		3 (15.79%)	
61 and above	3 (23.08%)		2 (15.38%)				1 (7.69%)		7 (53.85%)		1 (7.69%)	
χ^2^ _(30, 488)_ = 92.19, *p* < 0.001.												
(**b**)												
**Age** (**Years**)	**Daily**		**Several Times a Week**		**Once a Week**		**Once in a While**		**Rarely**		**Never**	
	**Male**	**Female**	**Male**	**Female**	**Male**	**Female**	**Male**	**Female**	**Male**	**Female**	**Male**	**Female**
1–10	11 (44.00)	16 (44.44)	2 (8.00)	6 (16.67)	0	2 (5.56)	2 (8.00)	1 (2.78)	3 (12.00)	2 (5.56)	7 (28.00)	9 (25.00)
11–20	7 (31.82)	19 (38.00)	3 (13.64)	7 (14.00)	4 (18.18)	10 (20.00)	6 (27.27)	5 (10.00)	2 (9.09)	5 (10.00)	0	4 (8.00)
21–30	30 (50.00)	20 (24.10)	6 (10.00)	13 (15.66)	10 (16.67)	19 (22.89)	8 (13.33)	23 (27.71)	4 (6.67)	5 (6.02)	2 (3.33)	3 (3.61)
31–40	13 (25.49)	6 (10.53)	15 (29.41)	9 (15.79)	10 (19.61)	8 (14.04)	6 (11.76)	16 (28.07)	1 (1.96)	12 (21.05)	6 (11.76)	6 (10.53)
41–50	7 (20.00)	3 (12.50)	6 (17.14)	6 (25.00)	10 (28.57)	1 (4.17)	8 (22.86)	5 (20.83)	4 (11.43)	4 (16.67)	0	5 (20.83)
51–60	6 (40.00)	0	1 (6.67)	2 (50.00)	2 (13.33)	0	1 (6.67)	1 (25.00)	3 (20.00)	0	2 (13.33)	1 (20.83)
61 and above	3 (30.00)	0	1 (10.00)	1 (25.00)	0	0	1 (10.00)	0	4 (40.00)	3 (75.00)	1 (10.00)	0
χ^2^ _(5, 488)_ = 7.83, *p* = 0.17.												

**Table 9 healthcare-13-00458-t009:** (**a**)**.** Use of tobacco use among the study participants across various age groups. (**b**)**.** Male and female differences in the use tobacco use among the study participants across various age groups. The notations in the brackets represent the percentage of the population.

(**a**)				
**Age** (**Years**)	**No**		**Yes**	
1–10	62 (100.00%)			
11–20	74 (91.34%)		7 (8.64%)	
21–30	118 (81.94%)		26 (18.06%)	
31–40	79 (75.96%)		25 (24.04%)	
41-50	43 (71.67%)		17 (28.33%)	
51–60	13 (72.22%)		5 (27.78%)	
61 and above	9 (75.00%)		3 (25.00 %)	
χ^2^ _(6, 398)_ = 27.61, *p* < 0.001.				
(**b**)				
**Age** (**Years**)	**No**		**Yes**	
	**Male**	**Female**	**Male**	**Female**
1–10	25 (100)	37 (100)		
11–20	29 (90.63)	45 (91.84)	3 (9.38)	4 (8.16)
21–30	49 (79.03)	69 (84.15)	13 (20.97)	13 (15.85)
31–40	39 (79.59)	40 (72.73)	10 (20.41)	15 (27.27)
41–50	24 (68.57)	19 (76.00)	11 (31.43)	6 (24.00)
51–60	12 (75.00)	1 (50.00)	4 (25.00)	1 (50.00)
61 and above	4 (57.14)	5 (100.00)	3 (42.86)	0
χ^2^ _(1, 398)_ = 1.18, *p* = 0.28.				

**Table 10 healthcare-13-00458-t010:** (**a**). Self-rating of one’s overall oral health by the study participants across various age groups. (**b**). Male and female differences in self-rating of one’s overall oral health across various age groups. The notations in the brackets represent the percentage of the population.

(**a**)										
**Age** (**Years**)	**Excellent**		**Very Good**		**Good**		**Fair**		**Poor**	
1–10 *										
11–20 **			3 (9.09%)		17 (51.51%)		7 (21.21%)		6 (18.18%)	
21–30	2 (1.43%)		7 (5.00%)		41 (29.28%)		59 (42.14%)		31 (22.14%)	
31-40			7 (6.80%)		23 (22.33%)		40 (38.83%)		33 (32.03%)	
41–50			3 (5.36%)		13 (23.21%)		22 (39.29%)		18 (32.14%)	
51–60			1 (6.25%)		4 (25.00%)		8 (50.00%)		3 (1.75%)	
61 and above			2 (18.18%)		3 (27.27%)		3 (27.27%)		3 (27.27%)	
(**b**)										
**Age** (**Years**)	**Excellent**		**Very Good**		**Good**		**Fair**		**Poor**	
	**Male**	**Female**	**Male**	**Female**	**Male**	**Female**	**Male**	**Female**	**Male**	**Female**
1–10										
11–20			3 (21.43)	0	5 (35.71)	12 (63.16)	4 (28.57)	3 (15.79)	2 (14.29)	4 (21.05)
21–30	2 (3.64)	0	4 (7.27)	3 (3.61)	21 (38.18)	20 (24.10)	24 (43.64)	35 (42.17)	6 (10.91)	25 (30.12)
31–40			5 (10.20)	2 (3.70)	10 (20.45)	13 (24.07)	25 (51.02)	15 (27.78)	9 (18.37)	24 (44.44)
41–50			3 (9.09)	0	7 (21.21)	6 (26.09)	15 (45.45)	7 (30.43)	8 (24.24)	10 (43.48)
51–60			0	1 (25.00)	2 (16.67)	2 (50.00)	7 (58.33)	1 (25.00)	3 (25.00)	0
61 and above			1 (14.29)	1 (25.00)	3 (42.86)	0	3 (42.86)	0	0	3 (75.00)
χ^2^ _(3, 357)_ = 20.45, *p* < 0.001.										

* The young respondents could not rate their overall oral healthcare ** 49 participants in the age group of 11–20 did not respond to this question. age: χ^2^
_(15, 357)_ = 19.53, *p* = 0.19; chi-square test, Table 10a,b.

## Data Availability

The original contributions presented in this study are included in the article. Further inquiries can be directed at the corresponding author).

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
