# Peer review of "Oral Health and Hygiene Practices from Baramulla District, Jammu and Kashmir, India: A Questionnaire-Based Cross-Sectional Observational Survey"

_healthcare, 2025, doi:10.3390/healthcare13050458_

Round 1
Reviewer 1 Report
Comments and Suggestions for Authors
This study investigates oral health and hygiene practices among residents of Baramulla District, Jammu and Kashmir, India, using a questionnaire-based cross-sectional survey. The study surveyed 488 participants across different age groups and genders to assess their oral hygiene habits, including brushing frequency, tongue cleaning, and dental visits. The results highlight suboptimal oral hygiene practices, with many participants brushing only once a day and making dental visits primarily in response to pain or discomfort. The study provides valuable insights into oral health behaviors in a previously under-researched region and emphasizes the need for oral health education and policy interventions. Addressing the following concerns can improve the quality of the manuscript.
1. While the study includes 488 participants, it lacks a detailed discussion on how representative the sample is of the broader Baramulla population. Information on selection criteria and potential biases should be elaborated.
2. The study relies on self-reported data, which may introduce reporting bias. Including a clinical oral health assessment by dental professionals would enhance the reliability of findings.
3. The study provides a snapshot of oral health behaviors at a single point in time. A longitudinal approach would better capture changes in oral hygiene practices and the impact of interventions over time.
4. While basic descriptive statistics are presented, advanced statistical methods such as regression analysis could provide deeper insights into factors influencing oral hygiene behaviors.
5. The study briefly categorizes socioeconomic status but does not explore how financial constraints, education levels, and cultural beliefs shape oral health practices.
6. The study identifies poor oral hygiene habits but does not sufficiently discuss strategies for improvement, such as school-based oral health programs, community initiatives, or policy recommendations.
7. Although the study includes both male and female participants, it does not thoroughly analyze gender-specific trends in oral hygiene behaviors and potential underlying causes. This section's statistical analysis can be studied by referring to these two articles (PMID:38995226, PMID:39515566).
Author Response
Manuscript ID: healthcare-3427610
Type of manuscript: Brief Report
Title: Oral health and hygiene practices from Baramulla district, Jammu and Kashmir, India: A Questionnaire-Based cross-sectional cross Survey
Reviewer # 1: This study investigates oral health and hygiene practices among residents of Baramulla District, Jammu and Kashmir, India, using a questionnaire-based cross-sectional survey. The study surveyed 488 participants across different age groups and genders to assess their oral hygiene habits, including brushing frequency, tongue cleaning, and dental visits. The results highlight suboptimal oral hygiene practices, with many participants brushing only once daily and making dental visits primarily in response to pain or discomfort. The study provides valuable insights into oral health behaviors in a previously under-researched region and emphasizes the need for oral health education and policy interventions. Addressing the following concerns can improve the quality of the manuscript.
Response #: Thank you for your encouraging comments. We have revised the text according to the suggestions of all the reviewers and hope that the revised form of the manuscript is more acceptable.
Comment # 1: While the study includes 488 participants, it lacks a detailed discussion on how representative the sample is of the broader Baramulla population. Information on selection criteria and potential biases should be elaborated.
Response # 1: We thank the reviewer for the suggestion. Following this, we have now revised the manuscript. Baramulla has a population of 1,008,039 residents (Census of India, 2011) of which 53.03 % are males and 46.95 % are females. The study’s sample size calculation was performed using the sample size calculator from the Department of Quantitative Health Sciences, Cleaveland Clinic (https://riskcalc.org/samplesize/). Our sample size of 488 was conveniently above the minimum of 384 participants required to reach a 95 % confidence level and a 5 % margin of error as per the sample size calculation from the calculator. Also, the proportion of males (46.93 %) and females (53.07 %) adequately represent both genders. People present on the hospital premises during the study period who agreed to participate in the survey were included in the study (Page 3, L94-95, L134-135; Page 4, L145-150).
Comment # 2: The study relies on self-reported data, possibly introducing reporting bias. Including a clinical oral health assessment by dental professionals would enhance the reliability of findings.
Response # 2: Thank you. We agree. We would like to respectfully submit here that we cannot conduct the oral health assessment by dental professionals at this stage now. But we will certainly include this in our future studies.
Comment # 3: The study provides a snapshot of oral health behaviors at a single point in time. A longitudinal approach would better capture changes in oral hygiene practices and the impact of interventions over time.
Response # 3: Thank you. We appreciate your positive feedback. We will surely do longitudinal studies in the future and focus on looking at the impacts of oral health intervention programs over time. The current study aims to provide the baseline data on oral health hygiene conditions of the Baramulla population in Jammu and Kashmir, India.
Comment # 4: While basic descriptive statistics are presented, advanced statistical methods such as regression analysis could provide deeper insights into factors influencing oral hygiene behaviors.
Response # 4: Thank you for your comment. We have worked on the statistical methods and incorporated chi-square analysis to understand age and gender differences in oral hygiene practices and overall oral health. Since our primary objective was to generate baseline data on oral hygiene behaviour in the Baramulla district, regression analysis was unsuitable for our dataset since it analyses relationships between dependent and independent variables with a causal-effect relationship. The variables included in this study did not have a causal-effect relationship. Despite this fact, following the reviewer’s suggestion, we attempted to apply logit regression between self-reported oral health (as a dependent variable) and factors such as age, gender and oral hygiene practices (as independent variables) both on the overall and individual age-wise datasets. We found no significant association between any dependent and independent variables. Therefore, we did not find it relevant to include the results of regression analyses in the manuscript.
Comment # 5: The study briefly categorizes socioeconomic status but does not explore how financial constraints, education levels, and cultural beliefs shape oral health practices.
Response # 5: We would like to respectfully state here that the data on education levels and financial conditions were collected based on the responses given in the questionnaire on the spot. It was impossible to verify the information because most respondents did not carry their income or educational certificates. Therefore, we have not made a detailed analysis of the effect of financial and educational levels and have only reported it from a preliminary overall perspective. Further, we did not collect any data on cultural beliefs. However, as per official census 2011, Muslims are majority in Baramula constituting 95.15% of the population.
Comment # 6: The study identifies poor oral hygiene habits but does not sufficiently discuss strategies for improvement, such as school-based oral health programs, community initiatives, or policy recommendations.
Response # 6: Thank you for your comment. We have now incorporated strategies for improvement and policy recommendations in the discussion section (Page 17, L420-429).
Comment # 7: Although the study includes both male and female participants, it does not thoroughly analyze gender-specific trends in oral hygiene behaviors and potential underlying causes. This section's statistical analysis can be studied by referring to these two articles (PMID:38995226, PMID:39515566).
Response # 7: Thank you for your suggestion. We have now incorporated the results of chi-square analysis to understand the age and gender-specific trends in oral hygiene behaviours in the revised manuscript. Chi-square analyses revealed a significant difference in various oral hygiene behaviours and self-reported oral health across age and gender. Specifically, the oral hygiene behaviors of tooth brushing frequency in a day, the timing of tooth brushing, tongue cleaning, use of mouthwash or other cleaning aids, frequency of eating fast food, and use of tobacco were significantly different across various age brackets. Males and females significantly differed in the frequency of tooth brushing and overall self-reported oral health.
Reviewer 2 Report
Comments and Suggestions for Authors
This study provides valuable insights into the oral health practices of the Baramulla district population in Jammu and Kashmir, focusing on demographic data, toothbrushing habits, mouthwash use, tongue cleaning, diet, and tobacco use. While this information offers a perspective on oral health behaviors, it falls short of linking these behaviors to actual oral health outcomes.
The assessment conducted by the authors relies solely on subjective evaluation through questionnaires. To provide a more comprehensive understanding, objective evaluation of the oral cavity’s condition, including the oral mucosa, teeth, and gums, is necessary through direct examination or screening of participants. This additional data would help establish whether the observed behaviors correlate with actual oral health status, thereby strengthening the study's findings and implications.
Author Response
Manuscript ID: healthcare-3427610
Type of manuscript: Brief Report
Title: Oral health and hygiene practices from Baramulla district, Jammu and Kashmir, India: A Questionnaire-Based cross-sectional cross Survey
Reviewer # 2: This study provides valuable insights into the oral health practices of the Baramulla district population in Jammu and Kashmir, focusing on demographic data, toothbrushing habits, mouthwash use, tongue cleaning, diet, and tobacco use. While this information offers a perspective on oral health behaviors, it falls short of linking these behaviors to actual oral health outcomes.
The assessment conducted by the authors relies solely on subjective evaluation through questionnaires. To provide a more comprehensive understanding, objective evaluation of the oral cavity’s condition, including the oral mucosa, teeth, and gums, is necessary through direct examination or screening of participants. This additional data would help establish whether the observed behaviors correlate with actual oral health status, thereby strengthening the study's findings and implications.
Response #: We thank the reviewer for finding merit in the submitted manuscript and the highly insightful comments. We primarily designed this study with the aim of collecting baseline data on the oral hygiene behaviors of residents of the Baramulla district. The study area is relatively unreported in scientific literature, and there is a vast literature gap on the oral hygiene and health of the study area populations. To the best of our knowledge, this study is the first one to report the oral hygiene conditions of the Baramulla residents, and only a couple of such studies exist from the Union Territory of Jammu and Kashmir. In future studies, we will surely incorporate all the suggestions on the objective evaluation and analyze the reasons behind the oral health outcomes in this manuscript. We have revised our manuscript in light of all the reviewers' comments and hope the revised form is more acceptable.
Reviewer 3 Report
Comments and Suggestions for Authors
This study reports survey results regarding basic oral health practices of a single institution in a specific region in India. While the topic of oral health is important, particularly in context of SDGs, this study does not make a meaningful contribution to the global or even national literature. It is well established that oral health practices can be improved in many developing regions of the world. While the survey results confirm this, the findings have minimal external validity. In fact, it may be hard to even extrapolate the result of this study to the rest of the region let alone the country of India as a whole. In regards to the methodology, the participants were selected at random, based on who walked into the clinic and who was willing to participate. There is no correlated objective clinical outcome measures to the self reported oral hygiene practices (other than self reported state of oral health). Further, there was no statistical analyses of the results, and no real conclusions made from the results, other than the fact that oral health practices need improvement.
In terms of improvement, it may be interesting to intervene on the findings of this survey and systematically measure outcomes in this group to see how interventions affect them, or even particular subgroups, such as the younger groups who eat more fast food. While the authors tackle an important issue in oral health, the results as reported currently simply do not add anything new that is not already in conventional knowledge. I would encourage the authors to continue with these endeavors and report more specific outcomes, particularly from any interventions that they may be able to enact.
Author Response
Manuscript ID: healthcare-3427610
Type of manuscript: Brief Report
Title: Oral health and hygiene practices from Baramulla district, Jammu and Kashmir, India: A Questionnaire-Based cross-sectional cross Survey
Reviewer # 3: This study reports survey results regarding basic oral health practices of a single institution in a specific region in India. While the topic of oral health is important, particularly in context of SDGs, this study does not make a meaningful contribution to the global or even national literature. It is well established that oral health practices can be improved in many developing regions of the world. While the survey results confirm this, the findings have minimal external validity. In fact, it may be hard to even extrapolate the result of this study to the rest of the region let alone the country of India as a whole. In regards to the methodology, the participants were selected at random, based on who walked into the clinic and who was willing to participate. There is no correlated objective clinical outcome measures to the self reported oral hygiene practices (other than self reported state of oral health). Further, there was no statistical analyses of the results, and no real conclusions made from the results, other than the fact that oral health practices need improvement.
In terms of improvement, it may be interesting to intervene on the findings of this survey and systematically measure outcomes in this group to see how interventions affect them, or even particular subgroups, such as the younger groups who eat more fast food. While the authors tackle an important issue in oral health, the results as reported currently simply do not add anything new that is not already in conventional knowledge. I would encourage the authors to continue with these endeavors and report more specific outcomes, particularly from any interventions that they may be able to enact.
Response #: Thank you for your comments. We would like to respectfully state here that this study aims to provide baseline data on the oral hygiene behaviors of residents of the Baramulla district. This study can serve as a baseline against which future studies can be planned to analyze the effect of different correlates and confounding factors on the oral health hygiene of residents of the study area. We have tried revising the manuscript in line with the reviewer’s comments. We have worked on the statistical methods and employed chi-square analysis to understand the effects of age and gender on oral hygiene behaviors. Although we cannot incorporate objective evaluation and analyze the reasons behind the oral health outcomes in this manuscript, we will surely include the suggested points in future studies. Also, intervention studies would be very important for understanding the impacts of oral health programs, particularly for young people. We hope that the revised form of the manuscript is more acceptable.
Reviewer 4 Report
Comments and Suggestions for Authors
Dear Authors,
I find your manuscript entitled “Oral Health and Hygiene Practices from Baramulla District, Jammu and Kashmir, India: A Questionnaire-Based Cross-Sectional Cross Survey” interesting and I consider that it could represent the starting point of more detailed research, thus bringing a contribution to the improvement of oral health status of the population in this region.
However, the paper has many points that need improvement. Please find below my observations and recommendations.
The Title section:
-“Cross-Sectional Cross Survey” - the term “cross” is repeated, please consider “Cross-Sectional Observational Survey” instead, or “Cross-Sectional Survey” and include statistical analysis
The Abstract section:
-this section should be structured with headings, as recommended in the “Instructions for Authors” on Journal’s site;
-please include some numerical findings of the study
The Introduction section:
-the total population of the district must be mentioned, as the study sample size depends on it.
The Methods section:
Subsection Study Subjects and Demography:
-please provide the number of ethical clearance from the institutional ethics committee;
-the selection of the study sample is not clear, it is mentioned that “it was not restricted to the patients who visited the hospital” (lines 119-120)- please be more specific;
-how was the sample size calculated? Please mention this;
- how did you ensure that the sample studied was representative of the total population of the district, in terms of age, gender, and socioeconomic distribution? In case the sample was not representative of the total population, the title of the manuscript should be changed.
Subsection Assessment through questionnaire:
-was the questionnaire an original one, or did you use a questionnaire previously used, from the literature? If the questionnaire was original, please provide the necessary details concerning its validation process: the pilot study, assessment of psychometric properties, Cronbach alpha coefficient. If it was taken from the literature, please provide details concerning the validation process in the study sample;
-please mention the necessary time for questionnaire completion;
-more questions would be useful concerning the diet (sugar intake) of the study population, and the use of fluoride products (according to the concentration of fluoride in the drinking water of the region).
Subsection Statistical analysis:
-line 158 – “significance” – what exactly does this mean? No statistical tests were used in the study, meaning that significance could not be assessed.
The Results section:
-Figure 1 showing the map of the study area should be moved to the Introduction or Methods section, it is not a finding of the present research. Also, the source of the image should be mentioned;
-Tables 3a and 3b should be placed after the paragraph citing them, not before it;
-subsection 3.1 is incorrectly entitled Oral health and hygiene, “oral health” means that results should be presented upon oral health status of the subjects. Please consider changing this subtitle into Oral hygiene behavior, or better dividing this section into multiple parts, according to the aspects assessed by the questions: Oral hygiene behavior, Oral health-related dietary behavior, Dental visits, Tobacco consumption, and Self-rating of oral health;
-Table 3b is not cited in text, please correct this;
-please express the results presented in Tables 3b, 4b, 5b, 6b, 7b, 8b, 9b, and 10b as percentages;
-Table 7c – please include the data concerning “pain and sensitivity”, for a clearer presentation of the results, and express the results as percentages;
-line 234 – the percentages mentioned cannot be found in Table 8a; please check and correct;
-Table 9b is not cited in text, please correct this;
-the expression “any deleterious habits like tobacco use” is imprecise – were other habits besides smoking evaluated?
- no results are presented at all according to socioeconomic status and educational level, although these would have been more useful than presenting comparative results according to gender, as they would allow pertinent conclusions to be drawn regarding the population groups on which oral health programs should be focused;
- a simple statistical analysis of the results would be very useful, however, and would add significance to the findings. This can be done by applying simple statistical tests to compare the data according to the variables included in the study: age, gender, socioeconomic level, educational level.
The Discussion section:
-more comparisons should be included with the results of similar studies conducted in other parts of India or in the world, by including numerical data, not only by mentioning “similar to studies”;
-lines 280-282 – please rephrase and be more specific: indicate the country/area of comparison, and the values found by these studies;
-please include more comments upon the study results.
The Conclusions section:
-this section should NOT contain references citations or comparisons with other studies or countries;
-please reformulate this section and refer strictly to the findings of the present research.
The manuscript is registered as a Brief Report, but it does not meet the requirements on the journal's website. I recommend that you complete the study with the aspects I presented above and classify it as a Research Article, or leave it as a Brief Report, but modify it so that it meets the necessary requirements.
Comments on the Quality of English LanguageRespected Editors,
In the manuscript entitled “Oral Health and Hygiene Practices from Baramulla District, Jammu and Kashmir, India: A Questionnaire-Based Cross-Sectional Cross Survey” the Authors present the results of an observational study conducted to fill the literature gap regarding Baramulla's oral health status and to understand the public's practices regarding the oral health and hygiene of the residents.
I find the manuscript interesting, and I consider that it could represent the starting point of more detailed research, thus bringing a contribution to the improvement of oral health status of the population in this region.
However, the paper has many points that need improvement. Please find below my observations and recommendations made to the Authors.
The Title section:
-“Cross-Sectional Cross Survey” - the term “cross” is repeated, the Authors should consider “Cross-Sectional Observational Survey” instead, or “Cross-Sectional Survey” and include statistical analysis
The Abstract section:
-this section should be structured with headings, as recommended in the “Instructions for Authors” on Journal’s site;
-some numerical findings of the study should be included.
The Introduction section:
-the total population of the district must be mentioned, as the study sample size depends on it.
The Methods section:
Subsection Study Subjects and Demography:
-the number of ethical clearance from the institutional ethics committee must be provided;
-the selection of the study sample is not clear, it is mentioned that “it was not restricted to the patients who visited the hospital” (lines 119-120)- the Authors should be more specific;
-how was the sample size calculated? This must be mentioned;
- Authors should also mention how they ensured that the sample studied was representative of the total population of the district, in terms of age, gender, and socioeconomic distribution. In case the sample was not representative of the total population, the title of the manuscript should be changed.
Subsection Assessment through questionnaire:
-more details are necessary: was the questionnaire an original one, or did they use a questionnaire previously used, from the literature? If the questionnaire was original, the necessary details concerning its validation process must be provided: the pilot study, assessment of psychometric properties, Cronbach alpha coefficient. If it was taken from the literature, details concerning the validation process in the study sample must be provided;
-the necessary time for questionnaire completion must be mentioned;
-more questions would be useful concerning the diet (sugar intake) of the study population, and the use of fluoride products (according to the concentration of fluoride in the drinking water of the region).
Subsection Statistical analysis:
-line 158 – “significance” –it is unclear what it means. No statistical tests were used in the study, meaning that significance could not be assessed.
The Results section:
-Figure 1 showing the map of the study area should be moved to the Introduction or Methods section, it is not a finding of the present research. Also, the source of the image should be mentioned;
-Tables 3a and 3b should be placed after the paragraph citing them, not before it;
-subsection 3.1 is incorrectly entitled Oral health and hygiene, “oral health” means that results should be presented upon oral health status of the subjects. The Authors should consider changing this subtitle into Oral hygiene behavior, or better dividing this section into multiple parts, according to the aspects assessed by the questions: Oral hygiene behavior, Oral health-related dietary behavior, Dental visits, Tobacco consumption, and Self-rating of oral health;
-Table 3b is not cited in text;
-the results presented in Tables 3b, 4b, 5b, 6b, 7b, 8b, 9b, and 10b should be expressed as percentages;
-Table 7c – the data concerning “pain and sensitivity” should be included, for a clearer presentation of the results, and the results should be expressed as percentages;
-line 234 – the percentages mentioned cannot be found in Table 8a; the Authors should check and correct;
-Table 9b is not cited in text;
-the expression “any deleterious habits like tobacco use” is imprecise – were other habits besides smoking evaluated? The Authors must clarify this;
- no results are presented at all according to socioeconomic status and educational level, although these would have been more useful than presenting comparative results according to gender, as they would allow pertinent conclusions to be drawn regarding the population groups on which oral health programs should be focused;
- a simple statistical analysis of the results would be very useful, however, and would add significance to the findings. This can be done by applying simple statistical tests to compare the data according to the variables included in the study: age, gender, socioeconomic level, educational level.
The Discussion section:
-more comparisons should be included with the results of similar studies conducted in other parts of India or in the world, by including numerical data, not only by mentioning “similar to studies”;
-lines 280-282 – the paragraph should be rephrased and the Authors should be more specific: indicate the country/area of comparison, and the values found by these studies;
-more comments upon the study results must be included.
The Conclusions section:
-this section should NOT contain references citations or comparisons with other studies or countries;
-the Authors should reformulate this section and refer strictly to the findings of the present research.
The cited references are appropriate and recent enough.
The manuscript is registered as a Brief Report, but it does not meet the requirements on the journal's website. I recommend that the Authors complete the study with the aspects I presented above and classify it as a Research Article, or leave it as a Brief Report, but modify it so that it meets the necessary requirements.
Author Response
Manuscript ID: healthcare-3427610
Type of manuscript: Brief Report
Title: Oral health and hygiene practices from Baramulla district, Jammu and Kashmir, India: A Questionnaire-Based cross-sectional cross Survey
Reviewer # 4:
Dear Authors,
I find your manuscript entitled “Oral Health and Hygiene Practices from Baramulla District, Jammu and Kashmir, India: A Questionnaire-Based Cross-Sectional Cross Survey” interesting, and I consider that it could represent the starting point of more detailed research, thus bringing a contribution to the improvement of oral health status of the population in this region.
However, the paper has many points that need improvement. Please find below my observations and recommendations.
Response #: We thank the reviewer for finding merit in our submitted manuscript. We have revised the manuscript by incorporating all the reviewers' suggestions and hope that the revised form is more acceptable.
The Title section:
Comment # 1: -“Cross-Sectional Cross Survey” - the term “cross” is repeated, please consider “Cross-Sectional Observational Survey” instead, or “Cross-Sectional Survey” and include statistical analysis
Response # 1: Thank You. We have corrected the title as suggested. The new title is “Oral Health and Hygiene Practices from Baramulla District, Jammu, and Kashmir, India: A Questionnaire-Based Cross-Sectional Observational Survey.”
The Abstract section:
Comment # 2: -this section should be structured with headings, as recommended in the “Instructions for Authors” on Journal’s site;-please include some numerical findings of the study
Response # 2: Thank You. We have structured the abstract with the headings as suggested and incorporated our study's quantitative findings in the abstract.
The Introduction section:
Comment # 3: -the total population of the district must be mentioned, as the study sample size depends on it.
Response # 3: We thank the reviewer for this insightful suggestion. We have now included the details of the population of Baramulla as such in the revised manuscript:
Page 3, L94-95: “Baramulla is the fourth most populous town in Jammu and Kashmir, with a population of 1,008,039 residents (Census of India, 2011) of which 53.03 % are males and 46.95 % are females.
The Methods section:
Subsection Study Subjects and Demography:
Comment # 4: -please provide the number of ethical clearance from the institutional ethics committee;
Response # 4: Thank You. As suggested, we have incorporated the details in the main text as following:
Page 4, L 142-144: “The study was conducted after seeking approval from the Office of the Medical Superintendent, Government Medical College, and its Associated Hospital, Baramulla (AH-GMCB/2023/907; 8th September 2023).”
Comment # 5: -the selection of the study sample is not clear, it is mentioned that “it was not restricted to the patients who visited the hospital” (lines 119-120)- please be more specific;
Response # 5: Thank You. We have revised the text. The study was done on the hospital premises, but the idea was to collect responses from the general population of the Baramulla district. We wanted to collect baseline data on the oral hygiene behaviors of the residents. Therefore, we did not just interview the patients who had visited the doctor for oral health issues but also interviewed people who had come to the premises with the patients or were present during the study. We have revised the text in the manuscript: “Participation in the study was voluntary, with proper ethics maintained. People present on the hospital premises during the study period who agreed to participate in the survey were included in the study.” (Page 3, L 143-137).
Comment # 6: -how was the sample size calculated? Please mention this;
Response # 6: Thank you for the comment. As suggested, we have now mentioned the details of the sample size calculation as follows in the revised manuscript:
Page 4, L 145-150: The study’s sample size calculation was performed using the sample size calculator from the Department of Quantitative Health Sciences, Cleaveland Clinic (https://riskcalc.org/samplesize/). The total population of Baramulla (as reported by Census of India, 2011) was 1,008,039. Our sample size of 488 was conveniently above the minimum of 384 participants required to reach a 95 % confidence level and a 5 % margin of error as per the sample size calculation from the calculator.
Comment # 7: - how did you ensure that the sample studied was representative of the total population of the district, in terms of age, gender, and socioeconomic distribution? In case the sample was not representative of the total population, the title of the manuscript should be changed.
Response # 7: Thank you for your comment. We would like to respectfully clarify here that for our study, we did convenience sampling; therefore, whoever was willing to participate was included. In terms of age, we divided our population into seven age groups. We had a fair distribution of our participants. More than 10 % of respondents were in all age groups except for 51 years and above. This is also because of the unwillingness of the older people to participate in the survey. Regarding gender, since we had more than one-third of our sample in each category (male and female), we felt it to be a good representative of the total population. Similarly, for socio-economic distribution, both the categories of BPL (below the poverty line) and APL (above the poverty line) had more than one-third of our sample (BPL: 41.39 %; and APL: 42.62 %), indicating a fair representation of the total population. The third socio-economic category, PHH/ AAY(15.98 %) are a special category of households that receive additional subsidies and food grains and are considered the poorest of the poor. We believe our sample was fair and representative of age, gender, and socio-economic distribution.
Subsection Assessment through questionnaire:
Comment # 8 -was the questionnaire an original one, or did you use a questionnaire previously used, from the literature? If the questionnaire was original, please provide the necessary details concerning its validation process: the pilot study, assessment of psychometric properties, Cronbach alpha coefficient. If it was taken from the literature, please provide details concerning the validation process in the study sample;
Response # 8: We would like to humbly clarify here that the questionnaire used in this study was based on already available literature from the World Health Organization. We based it from “Oral health surveys: basic methods”- 5th edition (https://iris.who.int/bitstream/handle/10665/97035/9789241548649_eng.pdf?sequence=1#page=66.15) (Page 4, 164-165). However, we would also like to state here that no validation was performed because we primarily aimed to collect the baseline data on the oral hygiene conditions of the respondents.
Comment # 8: -please mention the necessary time for questionnaire completion;
-more questions would be useful concerning the diet (sugar intake) of the study population, and the use of fluoride products (according to the concentration of fluoride in the drinking water of the region).
Response # 8: the study was conducted from September 2023-November 2023. (Page 3, L 121-122). Thank you for your suggestion on additional questions concerning diet, sugar intake and use of fluoride products. We will incorporate them in future studies.
Subsection Statistical analysis:
Comment # 9: -line 158 – “significance” – what exactly does this mean? No statistical tests were used in the study, meaning that significance could not be assessed.
Response # 8: We have revised the statistical analysis portion and deleted the line no. 158 (Statistical analysis, Page 5, L182-188).
The Results section:
Comment # 10: -Figure 1 showing the map of the study area should be moved to the Introduction or Methods section, it is not a finding of the present research. Also, the source of the image should be mentioned;
Response # 10: We thank the reviewer for pointing this out. As suggested, we have moved the map of the study area to the Methods section. Also, we would like to clarify here that our figure is an original figure that we prepared in Arc GIS 9.3 software using Digital Elevation Models (DEMs.). We have mentioned these details in the revised legend of Figure 1 in the manuscript:
Figure 1. Map representing the native locations of the respondents included in the study. All the respondents belonged to the Baramulla district of Jammu and Kashmir, India. The map was prepared in Arc GIS 9.3 software using Digital Elevation Model (DEM) of the study area as a base.
Comment # 11: -Tables 3a and 3b should be placed after the paragraph citing them, not before it;
Response # 11: We have now placed Table 3a and Table 3b after the paragraph citing them.
Comment # 12: -subsection 3.1 is incorrectly entitled Oral health and hygiene, “oral health” means that results should be presented upon oral health status of the subjects. Please consider changing this subtitle into Oral hygiene behavior, or better dividing this section into multiple parts, according to the aspects assessed by the questions: Oral hygiene behavior, Oral health-related dietary behavior, Dental visits, Tobacco consumption, and Self-rating of oral health;
Response # 12: Thank you for your constructive comment. We have revised the result section as suggested.
Comment # 13: -Table 3b is not cited in text, please correct this;
Response # 13: Corrected. (Page 8, L233).
Comment # 14: -please express the results presented in Tables 3b, 4b, 5b, 6b, 7b, 8b, 9b, and 10b as percentages;
Response # 14: Results presented as percentages in Tables 3b, 4b, 5b, 6b, 7b, 8b, 9b and 10b as suggested.
Comment # 15: -Table 7c – please include the data concerning “pain and sensitivity”, for a clearer presentation of the results, and express the results as percentages;
Response # 15: Results presented as percentages in Table 7c as suggested.
Comment # 16: -line 234 – the percentages mentioned cannot be found in Table 8a; please check and correct;
Response # 16: We have corrected the percentages mentioned and revised the text (Page 13, L298).
Comment # 17: -Table 9b is not cited in text, please correct this;
Response # 17: Corrected.
Comment # 18: -the expression “any deleterious habits like tobacco use” is imprecise – were other habits besides smoking evaluated?
Response #18: Thank you for your comment. By using the phrase “any deleterious habits”, we meant primarily tobacco usage and alcohol consumption. However, we did not evaluate alcohol consumption since we were mindful of the fact that the majority of the population in Baramulla is from the Muslim community, where drinking alcohol is considered “haram” or forbidden. We have now revised the text and replaced “any deleterious habits like tobacco use” with “tobacco usage”.
Comment # 19: - no results are presented at all according to socioeconomic status and educational level, although these would have been more useful than presenting comparative results according to gender, as they would allow pertinent conclusions to be drawn regarding the population groups on which oral health programs should be focused;
Response # 19: We thank the reviewer for this suggestion. We would like to respectfully submit here that the data on the socioeconomic status and educational level were gathered based on the responses given in the questionnaire on the spot. Verifying the information presented in the responses was not possible as the majority of the respondents did not carry their income (BPL) or educational certificates. Therefore, we found it prudent to exclude a detailed analysis based on the socio-economic status and educational levels and only include them as a reporting figure.
Comment # 20: - a simple statistical analysis of the results would be very useful, however, and would add significance to the findings. This can be done by applying simple statistical tests to compare the data according to the variables included in the study: age, gender, socioeconomic level, educational level.
Response # 20: Thank you. We have now incorporated the results of chi-square analysis to understand the age and gender-specific trends in oral hygiene behaviours in the revised manuscript.
The Discussion section:
Comment # 21: -more comparisons should be included with the results of similar studies conducted in other parts of India or in the world, by including numerical data, not only by mentioning “similar to studies”;
Response # 21: Revised as suggested.
Comment # 22: -lines 280-282 – please rephrase and be more specific: indicate the country/area of comparison, and the values found by these studies;
Response # 22: Rephrased as suggested.
Comment # 23: -please include more comments upon the study results.
Response # 23: Thank you for your constructive comment. We have elaborated the study results as suggested (Page 16, L348-351, L362-369, L373-375, L394; Page 17, L395-400).
The Conclusions section:
Comment # 24: -this section should NOT contain references citations or comparisons with other studies or countries; please reformulate this section and refer strictly to the findings of the present research.
Response 24: Thank you for the suggestion. We have revised the conclusion section and removed the references, citations, and comparisons to other studies.
Comment # 25: The manuscript is registered as a Brief Report, but it does not meet the requirements on the journal's website. I recommend that you complete the study with the aspects I presented above and classify it as a Research Article, or leave it as a Brief Report, but modify it so that it meets the necessary requirements.
Response # 25: We thank the reviewer for the suggestion. We have revised the document as per the Reviewers’ suggestions and comments. We will leave it to the discretion of the Editor to kindly consider it as a Research Article.
Round 2
Reviewer 1 Report
Comments and Suggestions for Authors
The authors addressed all the concerns.
Author Response
Manuscript ID: healthcare-3427610
Type of manuscript: Brief Report
Title: Oral health and hygiene practices from Baramulla district, Jammu and Kashmir, India: A Questionnaire-Based cross-sectional cross Survey
Reviewer # 1: The authors addressed all the concerns.
Response #: Thank you for your comment. We deeply appreciate your feedback which helped improve the manuscript.
Reviewer 2 Report
Comments and Suggestions for Authors
The present manuscript is well written.
However, the author should pay special attention to the Results section. The number of subjects in each table appears inconsistent. For example, in Table 3a, the "1–10 years" group with a toothbrush frequency of once per day includes 42 respondents. However, in Table 3b, the same group is divided into 17 males and 27 females, totaling 44 respondents, which does not match. Please check all the table
Author Response
Manuscript ID: healthcare-3427610
Type of manuscript: Brief Report
Title: Oral health and hygiene practices from Baramulla district, Jammu and Kashmir, India: A Questionnaire-Based cross-sectional cross Survey
Reviewer # 2: The present manuscript is well written. However, the author should pay special attention to the results section. The number of subjects in each table appears inconsistent. For example, in Table 3a, the "1–10 years" group with a toothbrush frequency of once per day includes 42 respondents. However, in Table 3b, the same group is divided into 17 males and 27 females, totaling 44 respondents, which does not match. Please check all the table.
Response #: We thank the reviewer for finding merit in the revised manuscript and their constructive comments. We have checked the results sections again and have corrected the error. We also thank the reviewer for the highly useful comments and suggestions in the previous rounds that greatly helped to improve the manuscript.
Reviewer 3 Report
Comments and Suggestions for Authors
I appreciate the authors' point that this is a baseline study. The addition of figure 1 and attempts at statistical analyses are positive additions to the manuscript. However, I believe the authors could benefit from statistical expertise consultation to help make edits throughout the manuscript. A few thoughts as below:
- For the methods section, there must be additional description that explicitly states what the primary predictor and primary outcome variables are, and the rationale for picking them.
- chi squared can be used for multi-level variables, but the authors should be very careful not to extrapolate significant p-values for pairwise comparisons. They certainly should not infer any casual relationship. The manuscript should be careully edited to remove any mention of causal relationship from these analyses. For example, "there was an effect of age but not gender on...". These type of statements should be restated as "there was an association between...".
- The authors should include some mention of notable analyses as part of the abstract of the manuscript.
Author Response
Response to Reviewer's Comment
Manuscript ID: healthcare-3427610
Type of manuscript: Brief Report
Title: Oral health and hygiene practices from Baramulla district, Jammu and Kashmir, India: A Questionnaire-Based cross-sectional cross Survey
Reviewer # 3:
Comment # 1: I appreciate the authors' point that this is a baseline study. The addition of Figure 1 and attempts at statistical analyses are positive additions to the manuscript. However, I believe the authors could benefit from statistical expertise consultation to help make edits throughout the manuscript. A few thoughts as below:
Response # 1: We thank the reviewer for the highly constructive comments and finding merit in the revised manuscript. As per the suggestion, we did take the help of our colleague at the department who specializes in data analysis and incorporated his suggestions in the manuscript.
Comment # 2: For the methods section, there must be an additional description that explicitly states what the primary predictor and primary outcome variables are and the rationale for picking them.
Response # 2: Thank you. To the best of our understanding, such variables are needed in regression analysis where there is a causal-effect relationship. Since our study was an explorative one and the responses in the questionnaire were not collected from the point of view of deciphering the causal-effect relationship, we think that stating primary predictor and outcome variables would create confusion for the readers in the absence of regression analyses.
Comment # 3: chi-squared can be used for multi-level variables, but the authors should be very careful not to extrapolate significant p-values for pairwise comparisons. They certainly should not infer any causal relationship. The manuscript should be carefully edited to remove any mention of causal relationships from these analyses. For example, "there was an effect of age but not gender on...". These types of statements should be restated as "there was an association between...".
Response # 3: We thank the reviewer for correcting us. We have corrected the language associated with the chi-square result in the result section.
Comment # 4: The authors should include some mention of notable analyses as part of the abstract of the manuscript.
Response # 4: Thank you for your comment. We have mentioned chi-square analyses in the abstract section (Page no. 1, L16-17).
Reviewer 4 Report
Comments and Suggestions for Authors
Respected Authors,
Thank you for considering my recommendations in the revised version of your manuscript entitled “Oral Health and Hygiene Practices from Baramulla District, Jammu and Kashmir, India: A Questionnaire-Based Cross-Sectional Observational Survey”.
Some important points still need adjustment.
1. Validation of the questionnaire. Although it was based on already available literature from the World Health Organization, the following is specified in the WHO brochure: “Interviewing requires appropriate field training of survey staff and a pilot study of 15–20 subjects should be carried out prior to the actual survey for ensuring face validity and reliability.” (Page 73 of the brochure)
Even translated questionnaires or questionnaires already used in a different population still need validation in the target population of the study.
You can use the data already gathered to perform this validation.
2. The request to mention the necessary time for questionnaire completion referred to the number of minutes necessary to answer all the questions. This is a characteristic that is usually mentioned in case of questionnaires.
3. The chi-square value does not need to be mentioned in the text, only the p-value which should also be found in the corresponding Tables (a final row can be added).
4. The results of the statistical analysis comparing the data by age and gender should be placed in the text in each subsection after the paragraph containing the data obtained for the entire study group.
5. The phrase between lines 283 and 274 should be moved to subsection 3.4.
6. The Discussion section still needs more comparisons with similar data from other studies in literature. The study that you conducted was a large one, and a lot of data was issued, requiring more detailed comments and comparisons.
7. The Conclusions section should be reformulated so that it refers concretely to the results of the present study and includes only a few remarks concerning the necessary actions to improve the situation that was found.
Comments on the Quality of English LanguageThe English could be improved to more clearly express the research.v
Author Response
Response to Reviewer's Comment
Manuscript ID: healthcare-3427610
Type of manuscript: Brief Report
Title: Oral health and hygiene practices from Baramulla district, Jammu and Kashmir, India: A Questionnaire-Based cross-sectional cross Survey
Reviewer # 4:
Respected Authors,
Thank you for considering my recommendations in the revised version of your manuscript entitled “Oral Health and Hygiene Practices from Baramulla District, Jammu and Kashmir, India: A Questionnaire-Based Cross-Sectional Observational Survey.” Some important points still need adjustment.
Response # We thank the reviewer for the constructive comments, which have immensely helped to improve the manuscript. We have further revised the manuscript in light of the new comments of the 4 reviewers and hope that the manuscript is more acceptable in the revised form.
Comment # 1: Validation of the questionnaire. Although it was based on already available literature from the World Health Organization, the following is specified in the WHO brochure: “Interviewing requires appropriate field training of survey staff and a pilot study of 15–20 subjects should be carried out before the actual survey ensuring face validity and reliability.” (Page 73 of the brochure). Even translated questionnaires or questionnaires already used in a different population still need validation in the target population of the study.
You can use the data already gathered to perform this validation.
Response # 1: We thank the reviewer for this useful suggestion. We do appreciate that we should have conducted validity of the questionnaire after the pilot study using standard measures such as Cronbach alpha as the reviewer himself/herself suggested in the last round. This is one of the major limitation of the study. As per the suggestion, we used the gathered data to perform Cronbach alpha validation in R software using “psy” package. But our Cronbach alpha was quite low (0.3). This was because our responses did not follow the same construct (scale of measurement). For example, the responses of “frequency of toothbrushing” was collected in the scale of 1-4, while the responses of “tongue cleaning” was collected in the scale of 1-2, “frequency of fast food consumption” was collected in the scale of 1-5, etc. Therefore, there was little covariance in the datasheet for different responses. Having said this, we again appreciate the limitation pointed out by the reviewer. We have stated this limitation as follows in the revised manuscript now:
Page no. 15 and 16, L424-426: “The present study should have ideally included a pilot study with 15-20 respondents as per the WHO recommendations, and the responses should have been validated with standard statistical measures such as Cronbach’s alpha.”
We are presenting a brief report on oral health from Jammu and Kashmir that has not been explored yet. So, this brief report is aimed at being the baseline for further full-scale research in the domain. In future studies, we will incorporate validation through a pilot study. Once again, we are thankful to the reviewer for this.
Comment # 2: The request to mention the necessary time for questionnaire completion referred to the number of minutes necessary to answer all the questions. This is a characteristic that is usually mentioned in the case of questionnaires.
Response # 2: Thank you. We have mentioned the necessary time to complete the questionnaire in the manuscript, “It took 7-10 minutes per subject to complete the questionnaire. However, in the case of participants under 10 years of age, it took slightly more time to complete the questionnaire since the answers were provided by the parents and sometimes interrupted by the participants” Page no.4, L162-165).
Comment # 3: The chi-square value does not need to be mentioned in the text, only the p-value, which should also be found in the corresponding tables (a final row can be added).
Response # 3: Thank you. We have revised the text as suggested. We have removed the chi-square values from the text and added it as footnotes with the respective tables.
Comment # 4: The results of the statistical analysis comparing the data by age and gender should be placed in the text in each subsection after the paragraph containing the data obtained for the entire study group.
Response # 4: Thank you. We have revised the text as suggested.
Comment # 5: The phrase between lines 283 and 274 should be moved to subsection 3.4.
Response # 5: Thank you. We have moved the lines as suggested (Page no. 11, L305-306).
Comment # 6: The Discussion section still needs more comparisons with similar data from other studies in the literature. The study that you conducted was a large one, and a lot of data was issued, requiring more detailed comments and comparisons.
Response # 6: Thank you. We have revised our discussion section as suggested and added more references for more comparative discussions (Page no. 14, L354-361; Page no. 15, L386-387; Page no. 15, L393).
Comment # 7: The Conclusions section should be reformulated so that it refers concretely to the results of the present study and includes only a few remarks concerning the necessary actions to improve the situation that was found.
Response # 7: Thank you for your comment. We have rewritten the conclusion section. Hope it is acceptable in the revised manuscript (Page no.16, L445-456).